# One month convection timescale on the surface of a giant evolved star

Wouter Vlemmings[1 ✉], Theo Khouri[1], Behzad Bojnordi Arbab[1], Elvire De Beck[1] & Matthias Maercker[1]

The transport of energy through convection is important during many stages of stellar evolution[1,2], and is best studied in our Sun[3] or giant evolved stars[4]. Features that are attributed to convection are found on the surface of massive red supergiant stars[5–8]. Also for lower-mass evolved stars, indications of convection are found[9–13], but convective timescales and sizes remain poorly constrained. Models indicate that convective motions are crucial to produce strong winds that return the products of stellar nucleosynthesis into the interstellar medium[14]. Here we report a series of reconstructed interferometric images of the surface of the evolved giant star R Doradus. The images reveal a stellar disk with prominent small-scale features that provide the structure and motions of convection on the stellar surface. We find that the dominant structure size of the features on the stellar disk is $0.72 \pm 0.05$ astronomical units. We measure the velocity of the surface motions to vary between $-18$ and $+20$ km s$^{-1}$, which means that the convective timescale is approximately one month. This indicates a possible difference between the convection properties of low-mass and high-mass evolved stars.

The M-type asymptotic giant branch (AGB) star R Doradus (see also the first section of the Methods) was observed with the Atacama Large Millimeter/submillimeter Array (ALMA) in five epochs spread over 4 weeks between 2 July and 2 August 2023. The observations around 338 GHz were made using the longest available ALMA baselines. Using superuniform weighting, this allowed us to reconstruct images of the stellar surface at an angular resolution between 8 and 25 mas (see the first section of the Methods). At submillimetre wavelengths, the main opacity source is the electron-neutral free–free interactions[15]. As a result, the observations are sensitive to the temperature and density structure in an extended atmosphere above the stellar photosphere. In particular, submillimetre observations probe the dynamics and properties of the shocks excited by large convective cells on the stellar photosphere, while not being hindered by dust or molecular opacity sources that dominate at other wavelengths[11]. The surface maps for the three observational epochs with the highest angular resolution are shown in Fig. 1; the remaining epochs are shown in Extended Data Fig. 1. The maps show a predominantly circular stellar disk with a radius of $R_{338GHz} = 1.64 \pm 0.09$ astronomical units (AU) and a brightness temperature of $T_b = 2{,}270 \pm 130$ K (see the first section of the Methods). Considering the correspondence between several epochs, including observations taken at 225 GHz shown in the first section of the Methods, we conclude that the structures that we observe are intrinsic to the star and are probably induced by surface convection. From the visual inspection, we find that the structures we observe at the surface have a typical lifetime of at least 3 weeks.

To estimate the size of the dominant structures on the submillimetre surface, we use a spatial power spectral density (PSD) analysis (see the description in the first section of the Methods). The PSD analysis was carried out using the calibrated interferometric visibilities after subtracting the best fit model for the stellar disk in the three last, highest angular resolution, epochs. We show the results in Fig. 2. We find that structure exists at several scales on the surface. The structure varies between the three epochs, particularly in amplitude, and most of the power is concentrated in small-scale structure with a size of $x = 13.1 \pm 0.6$ mas. At a distance of R Doradus of $55 \pm 3$ pc (ref. 16), this corresponds to a size of $x = 0.72 \pm 0.05$ AU. Other peaks present in the power spectrum are located at $16.0 \pm 1.0$, $17.9 \pm 0.6$, $22.1 \pm 1.3$ and $35.2 \pm 3.6$ mas. The largest of these corresponds to more than half the visible hemisphere. The different scales might be related to the equivalent scales of granulation, mesogranulation and, considering the scales, in particular supergranulation inferred for our Sun[17,18], or could be the result of a superposition of several independent granules. In the absence of a better name, we subsequently use the term granules for the smallest structures, although concluding a direct link with Solar granules is not yet possible. On average, the power on each of the four larger scales is about 25% of that in the dominant structure.

The structures inside the stellar disk can also be compared with structures seen on the limb of the star. In Fig. 3a, we show the half-power radius as a function of position angle for the three highest angular resolution epochs. There is a good correspondence between the epochs, with specifically the last two epochs showing very similar structures in the half-power radius in Fig. 3a. As shown in Extended Data Fig. 4, also the 225 GHz observations show similar features, confirming that they are intrinsic to the star. By determining the difference in the radius between consecutive epochs at every position angle, we can derive the average velocity profile of the 338 GHz optical depth surface as shown in Fig. 3. Negative velocities indicate a radial motion inwards to the star. Although the average velocity is small, the movement of the optical depth surface as a function of angle varies between $V = -18$

[1]Department of Space, Earth and Environment, Chalmers University of Technology, Gothenburg, Sweden. ✉e-mail: wouter.vlemmings@chalmers.se

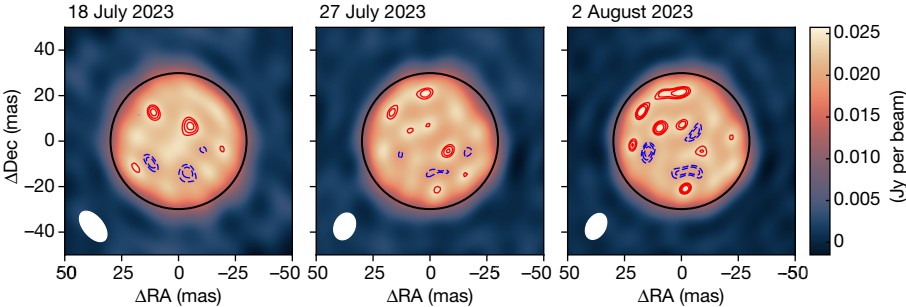

**Fig. 1 | The stellar surface of the AGB star R Doradus.** The panels represent the three highest angular resolution epochs of ALMA observations at 338 GHz. The black ellipse in the panels indicates the average size of the stellar disk at this frequency. The red solid contours and the blue dashed contours indicate the positive and negative 4, 5 and 6σ features with respect to the mean emission of the stellar disk. The size and orientation of the interferometric beam is indicated at the bottom left of each panel. DEC, declination; RA, right ascension.

and +20 km s$^{-1}$. These velocities can be compared with the local sound speed $V_s \approx 6$ km s$^{-1}$ in the part of the extended atmosphere probed by the observations and are consistent with supersonic shocks induced by convection (for example, ref. 19). Although the observed motions depend on a combination of changes in gas density, temperature, ionization and velocity, they are, on short timescales, a good representation of the (radial) motions of the shocks. They thus represent a lower limit to the actual gas velocities. The escape velocity at the measured submillimetre radius for a star with the parameters adopted for R Doradus is $V_{esc} = 30 \pm 3$ km s$^{-1}$. Hence, because of these shocks, only a very small fraction of the gas will be able to escape the gravity field of the star before being accelerated by radiation pressure on dust[14]. Considering the measurement of the size of the granules as well as the velocity of the shocks, we can determine the timescale for the surface structures to readjust (for example, ref. 20). This timescale is given by $t_{surf} \approx x/\Delta V = 33 \pm 3$ days, which is independent of the assumed distance to the star. The measurement of this timescale corresponds well with the timescale of more than 3 weeks estimated from the visual inspection of the images and is consistent with model predictions[21] but is only about 30% of the convective decay time extrapolated using the empirical formula (equation (16)) from ref. 19. As this empirical formula was derived for models of stars with both higher effective temperatures and surface gravity, our result points to a possible difference of convection properties for AGB stars.

The contrast between the average brightness of the stellar disk and the granules induced by the surface convective motions varies between $2.8 \pm 1.2$, $3.2 \pm 0.2$ and $1.5 \pm 0.2\%$ during the last three epochs (Fig. 2). This is notably less than the contrast of about 12% observed for the star π$^1$ Gruis in the near-infrared. As the contrast relates to the imprint of the convective motions on the shock structure in the extended atmosphere, we can use the contrast and size to determine the average brightness temperature increase $\Delta T_b$ caused by these shocks. We find $\Delta T_b$ to be between about 700 and 1,500 K. As the brightness temperature is closely related to the real temperature of the extended atmosphere (for example, refs. 11,15), this is sufficient to affect the chemistry in the stellar atmosphere (for example, ref. 22). Previous observations have also revealed brighter and more compact hotspots on the submillimetre surface of AGB stars, including R Doradus (for example, refs. 11,23,24). These hotspots had a contrast in excess of 5% that, because of their compact size (<0.3 AU), corresponded to an increase in brightness temperature that can reach $\Delta T_b > 50,000$ K. These hotspots have different characteristics from the convective shock structures described here and are rarer. Further monitoring observations are needed to determine the timescales related to the bright hotspots and characterize their origin. Although in the future, our observations can serve as a benchmark for the models of convection in M-type AGB stars, at present it is possible only to compare the results with parametric formulae that predict the granular size scale

on the basis of fundamental stellar properties such as effective temperature, surface gravity and chemical composition. All relations are based on mixing length theory and extrapolated models of less evolved stars, and there are no models yet that specifically aim to reproduce R Doradus. In Fig. 4, we compare our measurements (using the stellar parameters described in the first section of the Methods) with the three

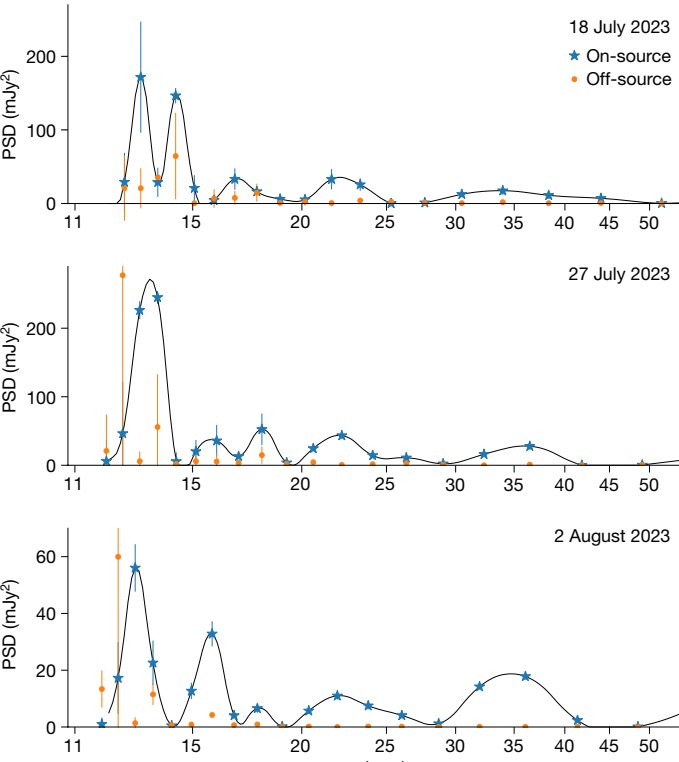

**Fig. 2 | The spatial PSD for three epochs of observations of R Doradus.** The spatial PSD, in units of mJy$^2$, was determined directly from the interferometric visibilities after subtracting the best fit model for the stellar disk. The three panels represent the final three, highest angular resolution, epochs. Note the nonlinear scale of the $x$-axis. The filled circles denote the measurements and the curve represents a cubic spline fit to the observations. The largest blue stars, with corresponding 1σ s.d. error bars, are the PSD determined on-source; the smaller orange dots represent the off-source measurements. In the last two epochs, the second bin at about 12 mas contains only limited visibilities and hence does not present a significant detection. As the peak at the smallest angular size dominates, this indicates that the smallest granule structure dominates. From an error-weighted average of the three epochs, we find a value of $x = 13.1 \pm 0.6$ mas. For a distance of $55 \pm 3$ pc (ref. 16), this corresponds to $x = 0.72 \pm 0.05$ AU, compared to a stellar diameter at 338 GHz of 3.28 AU.

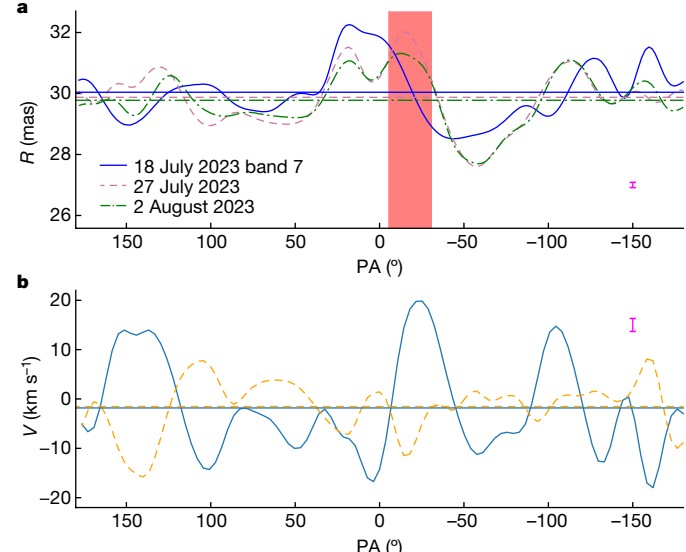

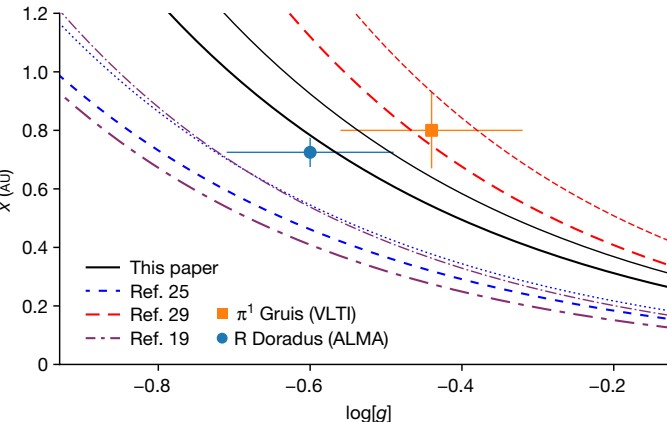

**Fig. 3 | The radius and radial velocity of the stellar surface of R Doradus.**
**a**, The half-power radius $R$ as a function of the position angle (PA) with respect to the north celestial pole. Positive angles point in the direction of the right ascension. The curves indicate $R$ measured for three epochs of ALMA band 7 (338 GHz) observations. The horizontal lines indicate the average radius for each epoch. The lower figure boundary on the radius is the estimated photospheric radius of 25.6 mas, measured at 2.3 μm (ref. 28). The red vertical bar indicates the granule size measured from the spatial PSD, with the linear size translated to an angular size at $R$. The $1\sigma$ s.d. on the radius (similar for each epoch) is plotted in the bottom right corner (magenta error bar) and is at most 0.17 mas. **b**, The radial velocity $V$ at the surface of R Doradus. The velocity is determined from the difference of the half-power radius between the third and fourth epochs (18 and 27 July 2023; solid line) and that between the fourth and fifth epochs (27 July and 2 August 2023; dashed line). The horizontal lines denote the average velocity. The velocity determined in this way corresponds to the movement of the $\tau_{338GHz} = 1$ optical depth surface and is an average between the respective epochs. This indicates the movement of the shocks induced by the convective motions. The $1\sigma$ s.d. in the radius determination translates to an uncertainty on the velocity of $\lesssim 2.6$ km s$^{-1}$ and is indicated in the top right of the panel (magenta error bar).

**Fig. 4 | Size of the smallest granules on the surface of R Doradus determined using ALMA compared with theoretical predictions and the previous Very Large Telescope Interferometer measurement for π¹ Gruis[10].** The observations are plotted versus log[$g$], and the $1\sigma$ s.d. error bars include the uncertainty on the distance and stellar mass. The solid black lines indicate the parametric scaling presented in this paper. The thick line was calculated for a stellar effective temperature $T_{eff} = 2,710$ K, which is the value for R Doradus. The thin line is the same model for $T_{eff} = 3,200$ K, which is closer to the value for π¹ Gruis. The other lines (dotted[25], dashed[29] and dash-dotted[19]) represent parametric models from the literature, with the thick and thin lines calculated for the stellar temperatures of 2,710 and 3,200 K, respectively. For the model from ref. 19, we assume solar metallicity. VLTI, Very Large Telescope Interferometer.

captured in a free parameter that has a large effect on galaxy evolution models (for example, ref. 27).

## Online content

relations that were also compared to the measurements of π¹ Gruis[10]. Although there is generally good agreement, none of the relations is a perfect fit. The size of the granulation is generally thought to scale with the pressure scale height $H_p$ immediately below the photosphere[4], with $x = \alpha H_p$. For AGB stars, the scaling parameter $\alpha$ is assumed to be about 10 (ref. 25). For the more massive red supergiant stars, $\alpha \approx 30$–50 seems to better match the hydrodynamical simulations and observations[8,26]. Using $H_p = k_B T/\mu g$, in which $k_B$ is the Boltzmann constant, $\mu$ is the mean molecular mass and $g$ is the acceleration due to surface gravity, this leads to the relation:

$$x = 0.427(\alpha/10)(T_{eff}/2,500 \text{ K})(g/0.25 \text{ cm s}^{-2})^{-1} \quad (1)$$

giving a result in astronomical units. Using the two AGB measurements, we can estimate a value of $\alpha = 17^{+5}_{-3}$ that represents the best fit to both observations.

Our results indicate that the convective structure on the surface of AGB stars matches the existing parametric formulae in size but reveal a difference between AGB stars and the more massive red supergiants. The results match the timescales found in AGB models, but the timescales differ from extrapolations of models based on less evolved stars and thus serve as a unique benchmark for the existing theory. The derived convection properties can also be implemented in stellar evolution and population synthesis models in which convection is generally

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

# Article

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

# Methods

## Source properties

R Doradus is an M-type AGB star that belongs to the class of semi-regular pulsators. On a timescale of about 1,000 days, it switches between two pulsation modes with periods of 362 and 175 days. The distance to R Doradus has been determined to be $55 \pm 3$ pc using revised Hipparcos measurements[6]. There is no usable Gaia parallax. From CO observations, it was determined that R Doradus has a relatively low mass loss rate (about $10^{-7} M_\odot$ yr$^{-1}$) and wind expansion velocity (about 5.7 km s$^{-1}$)[30]. Previous ALMA observations also indicated that R Doradus rotates fast for a giant star, with a rotation velocity at the surface of about $1.0 \pm 0.1$ km s$^{-1}$ compared to a rotation velocity of a few tens of metres per second expected for solitary AGB stars[31]. It has been suggested that the apparent rotation could be the result of a chance alignment of convective cells[20]. However, the rotation has been observed in several molecular lines at four different epochs that, including the observations presented here, span more than 6 years (for example, refs. 12,31). This is much longer than the convective timescales found in our analysis, and hence a chance alignment of convective cells can be ruled out.

In our comparison with the convective theory, we adopt the values for the effective temperature of $T_{eff} = 2{,}710 \pm 70$ K. For the surface gravity, we use $\log[g] = -0.6 \pm 0.1$, based on models that indicate that the initial mass was $1-1.25 M_\odot$ and that the current mass is $0.7-1.0 M_\odot$, combined with interferometric measurements that yield a stellar diameter in the infrared of $D_{IR} = 51.18 \pm 2.24$ mas (ref. 28). It is expected that this diameter, which corresponds to a radius of $R_{IR} = 1.4 \pm 0.1$ AU $= 298 \pm 21 R_\odot$, indicates the size of the stellar photosphere. We can compare this with the ($\tau = 1$) size of the star determined with ALMA at 338 GHz obtained by visibility fitting. Using a combination of the last three epochs, we fit a nearly circular stellar disk with $F_{338GHz} = 521 \pm 18$ mJy, $D_{338GHz} = 59.8 \pm 0.4$ mas and an axis ratio of $0.99 \pm 0.01$. This means a brightness temperature $T_b = 2{,}270 \pm 130$ K and, taking into account the uncertainty on the distance, a radius $R_{338GHz} = 1.64 \pm 0.09$ AU $= 353 \pm 19 R_\odot = 1.18 \pm 0.11 R_{IR}$.

## Observations, data reduction and imaging

The AGB star R Doradus was observed in ALMA bands 6 and 7 as part of the ALMA project 2022.1.01071.S (principal investigator: T.K.). The band 7 observations were taken between 5 July and 2 August 2023 using four spectral windows centred at 331.2, 333.0, 342.1 and 345.1 GHz. Each spectral window had a bandwidth of 1.875 GHz and 1,920 channels. The integration time of the individual visibilities was set to 2.02 s. The observations were taken in the largest ALMA configurations (C-9 and C-10) with the quasars J0519-4546 and J0516-6207 as bandpass–amplitude and phase calibrator, respectively. Details of the observations are presented in Extended Data Table 1. The calibration of the last three epochs was carried out using the ALMA pipeline in CASA v6.4.1.12[32]. The first two epochs were labelled as semi-pass in the ALMA quality assurance and for these the calibration was carried out manually by staff from the Nordic ALMA Regional Center node using CASA v6.5.4.9. In the first epoch, there was an issue with the bandpass calibration that needed to be solved manually. In the second epoch, one of the antennas needed to be flagged, resulting in a loss of some of the longest baselines. For both epochs, the requested angular resolution and sensitivity were not reached. After the initial calibration of each epoch, molecular lines were identified and flagged before the data were averaged to an integration time of 6.06 s and to 50 channels per spectral window. Subsequently, two steps of phase-only self-calibration were carried out on the stellar continuum. The self-calibration improved the signal-to-noise ratio by a factor of about 2.5 on the continuum. Finally, images, using all four spectral windows, were produced for the five epochs using superuniform visibility weighting[33]. This method increases the relative weight of the visibilities at the longer baselines, which minimizes the beam size at the expense of signal-to-noise ratio. The superuniform beam characteristics and continuum root mean square (r.m.s.) noise are also given in Extended Data Table 1. The increase of r.m.s. noise compared to more regular Briggs weighting (with a robust parameter of 0.5) depends on the telescope distribution and was a factor of about 2, 1.5, 1.5, 3 and 7 for the five epochs, respectively. The improvement in angular resolution between superuniform and uniform weighting ranged from about 2% in the final epoch to about 15% in the third epoch. The superuniform-weighted images of the final three epochs are presented in Fig. 1 and those of the first two epochs are shown in Extended Data Fig. 1. The images of the highest-angular-resolution epochs were used to derive the angular radial profile presented in Fig. 3. We verified that reducing the angular resolution to match that of the epoch with the largest beam smooths out the observed structures and the derived velocities and thus use the highest-angular-resolution results. The fits of the stellar disk and the spatial PSD analysis were carried out on the calibrated visibilities. Although we focus our analysis on the higher-resolution band 7 observations, we also include the continuum result from the band 6 observations in the first section of the Methods. The observational details for these observations are also included in Extended Data Table 1, and the calibration, self-calibration and imaging steps were identical to those carried out for the band 7 observations. The four spectral windows are centred at 218.9, 220.8, 230.0 and 232.9 GHz, and the increase in the continuum r.m.s. noise between Briggs weighting and superuniform weighting is a factor of 1.5.

## Spatial PSD analysis

The spatial PSD is regularly used to derive information about, for example, the turbulent structure of the interstellar medium (for example, refs. 34,35) as well as the convective structure of the solar photosphere (for example, ref. 36). The spatial PSD is given by the two-dimensional Fourier transform of an image. However, considering interferometric images are themselves the Fourier transform of the interferometric visibilities, the spatial PSD is equal to the modulus squared of the complex visibilities[34]. We can thus calculate the PSD directly from our interferometric visibilities without introducing potential artefacts during the imaging process. As the PSD would be dominated by the power at the scales of the stellar disk, we first use the $uv$-fitting code uvmultifit[37] to fit the stellar disk in $uv$-coordinates (for a discussion on the disk profile, see the first section of the Methods). Subsequently, we subtract the disk from the visibilities, after which, we calculate, for a phase centre towards the centre of the star, the PSD using visibilities annularly averaged in equally spaced bins of uv–distance (in units of k$\lambda$). From the inverse of the $uv$–distance, we can directly obtain the angular scale $x$ in milliarcseconds. In addition, we also determine the PSD for a position offset from the star. We chose an offset of 7 as to avoid a contribution to the off-source PSD from the source signal. As the first two epochs have worse spatial resolution, we produce the PSD only for the final three epochs. The results are shown in Fig. 2. By comparing the on- and off-source PSD, we can determine which structures are significant and which are possibly due to correlated noise in the visibilities.

## Stellar disk profile

To check how well a top-hat-shaped stellar disk profile fits the observations, we have also investigated the stellar disk profile in the image plane. We produced a radially averaged profile of R Doradus based on the combined data for the final three epochs. We then compared this profile with a top-hat-shaped stellar disk model convolved with the interferometric beam. The results of this comparison are shown in Extended Data Fig. 2. The stellar disk model can accurately describe the observations with residuals at a level of about 2% of the peak emission. This means that the 338 GHz optical depth $\tau_{338GHz}$ increases steeply over only a small change in the radius. The radial motions we observe thus reflect the physical motion of the 338-GHz optical depth surface. As the optical depth is a strong function of the density[11], the motions closely reflect the motion of the shocks induced by the convection.

# Article

## Band 6 observations

The ALMA band 6 (225 GHz) observations and the radius as a function of position angle are shown in Extended Data Figs. 3 and 4, respectively. Fitting the interferometric visibilities yields a completely circular stellar disk with $F_{225GHz} = 221.4 \pm 0.1$ mJy and $D_{225GHz} = 61.8 \pm 0.1$ mas. This means a brightness temperature $T_b = 2{,}006 \pm 8$ K and, including the distance uncertainty, a radius $R_{225GHz} = 1.70 \pm 0.09$ AU $= 365 \pm 20\, R_\odot = 1.22 \pm 0.11\, R_{IR}$. In a comparison between the radii of the 225-GHz observations and the last epochs of the 338-GHz observations in Fig. 4, it is clear that there is a very good correspondence. As the different observations are completely independent, this shows that the pattern seen in the radii is intrinsic to the source at the time of the observations.

## Data availability

The ALMA data are publicly available on the ALMA archive (https://almascience.eso.org/aq/) as part of project 2022.1.01071.S. The spatial PSD for the three epochs is available in the CSV file RDor-psd.csv. This constitutes the source data for Fig. 2. The half-power radius and velocity that are the source data for Fig. 3 are available in the CSV file RDor-radiusvelocity.csv. The radial profile of R Doradus from Extended Data Fig. 2 is provided in the CSV file RDor-profile.csv. Source data are provided with this paper.

## Code availability

This paper makes use of the National Radio Astronomy Observatory CASA package and Python. The code used for *uv*-fitting, uvmultifit, is publicly available[37].

30. Ramstedt, S. & Olofsson, H. The $^{12}$CO/$^{13}$CO ratio in AGB stars of different chemical type. Connection to the $^{12}$C/$^{13}$C ratio and the evolution along the AGB. *Astron. Astrophys.* **566**, 145 (2014).

31. Vlemmings, W. H. T. et al. Rotation of the asymptotic giant branch star R Doradus. *Astron. Astrophys.* **613**, L4 (2018).

32. Hunter, T. R. et al. The ALMA interferometric pipeline heuristics. *Publ. Astron. Soc. Pac.* **135**, 074501 (2023).

33. CASA Team. et al. CASA, the Common Astronomy Software Applications for radio astronomy. *Publ. Astron. Soc. Pac.* **134**, 114501 (2022).

34. Crovisier, J. & Dickey, J. M. The spatial power spectrum of galactic neutral hydrogen from observations of the 21-cm emission line. *Astron. Astrophys.* **122**, 282–296 (1983).

35. Green, D. A. A power spectrum analysis of the angular scale of Galactic neutral hydrogen emission towards L=140 deg, B=0 deg. *Mon. Not. R. Astron. Soc.* **262**, 327–342 (1993).

36. Wedemeyer-Böhm, S. & Rouppe van der Voort, L. On the continuum intensity distribution of the solar photosphere. *Astron. Astrophys.* **503**, 225–239 (2009).

37. Martí-Vidal, I., Vlemmings, W. H. T., Muller, S. & Casey, S. UVMULTIFIT: a versatile tool for fitting astronomical radio interferometric data. *Astron. Astrophys.* **563**, 136 (2014).

**Acknowledgements** W.V., T.K. and B.B. acknowledge support from the Olle Engkvist Foundation through grant number 229-0368 and from the Swedish Research Council under grant numbers 2020-04044 and 2019-03777. We also acknowledge support from the Nordic ALMA Regional Centre node based at Onsala Space Observatory. The Nordic ALMA Regional Centre node is financed through Swedish Research Council grant number 2019-00208. This paper makes use of the following ALMA data: ADS/JAO.ALMA#2022.1.01071.S. ALMA is a partnership of the European Organisation for Astronomical Research in the Southern Hemisphere (representing its member states), the National Science Foundation (USA) and the National Institutes of Natural Science (Japan), together with the National Research Council (Canada), the National Science Council and the Academia Sinica Institute of Astronomy and Astrophysics (Taiwan), and the Korea Astronomy and Space Science Institute (Republic of Korea), in cooperation with the Republic of Chile. The Joint ALMA Observatory is operated by the European Organisation for Astronomical Research in the Southern Hemisphere, Associated Universities, Inc. and the National Radio Astronomy Observatory, and the National Astronomical Observatory of Japan.

**Author contributions** W.V. wrote the paper and carried out the analysis of the data. T.K. wrote the observing proposal. W.V., T.K., B.B., E.D.B. and M.M. contributed to the interpretation of the results.

**Funding** Open access funding provided by Chalmers University of Technology.

**Competing interests** The authors declare no competing interests.

**Additional information**
**Correspondence and requests for materials** should be addressed to Wouter Vlemmings.

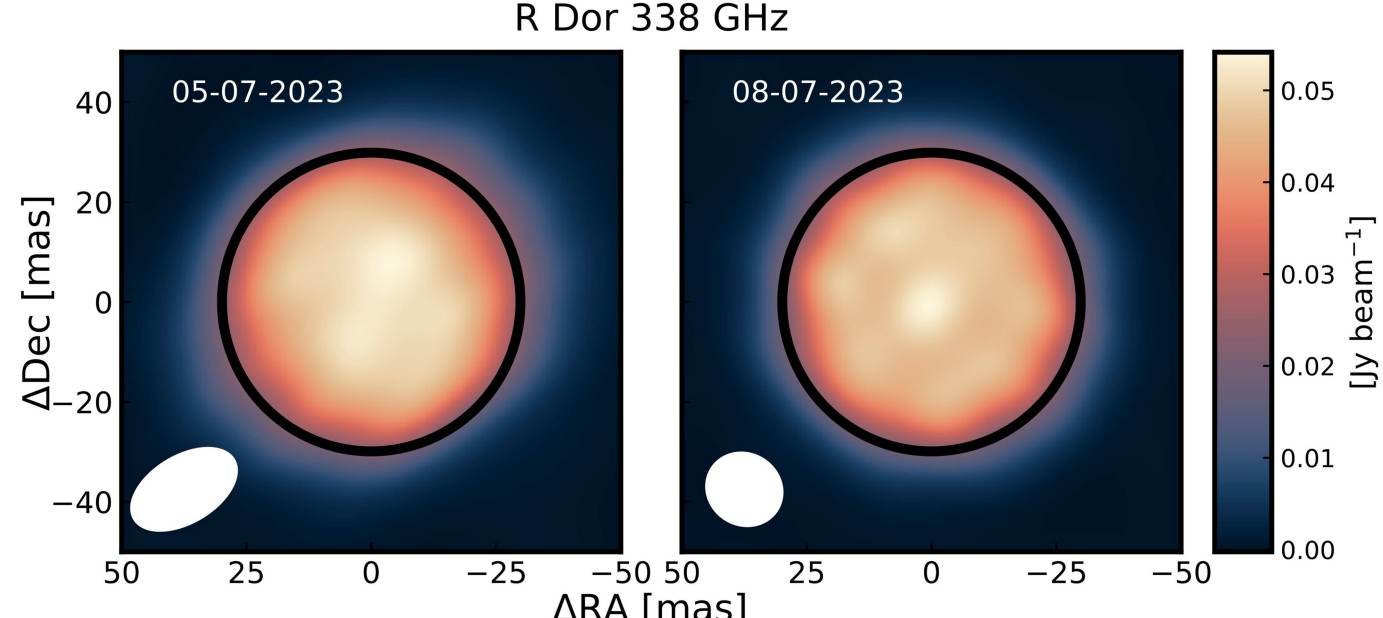

**Extended Data Fig. 1 | The stellar surface of the AGB star R Doradus.** The panels represent the first two epochs of ALMA observations at 338 GHz. The black ellipse in the panels indicates the average size of the stellar disc at this frequency. The size and orientation of the interferometric beam is indicated at the bottom left of each panel.

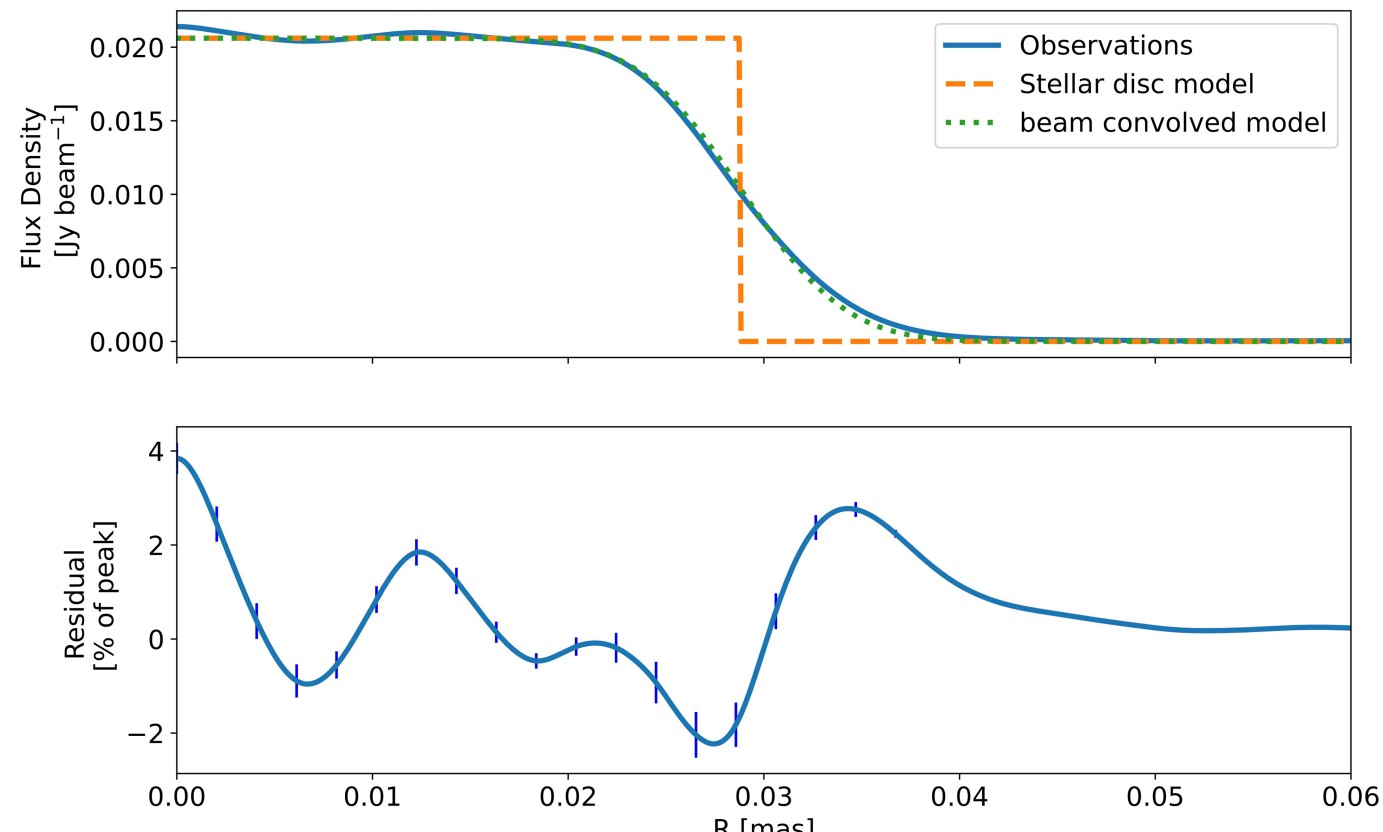

**Extended Data Fig. 2 | The observed azimuthally averaged radial profile of the stellar disc of R Doradus.** The radial profile, in the top panel, is generated from a combination of the three last epochs. The dashed line indicates the uniform disc model that minimizes the residual flux shown in the bottom panel. The error bars on the radial profile are close to the line width, with a largest error of $5 \times 10^{-5}$ Jy beam$^{-1}$. The error bars on the residual fraction are indicated with the vertical line segments.

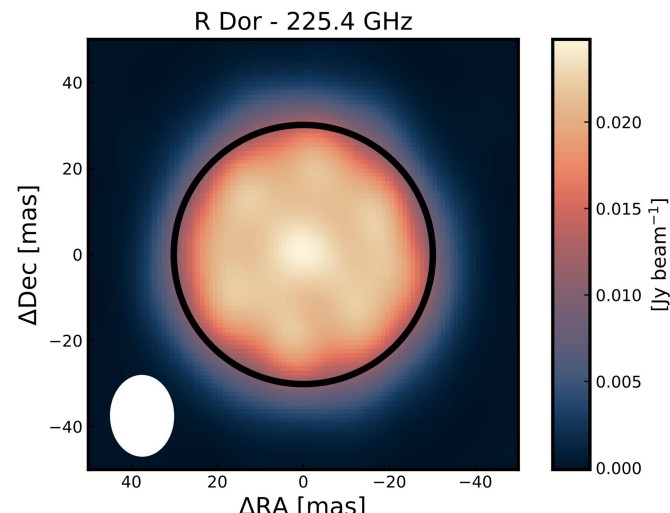

**Extended Data Fig. 3 | The stellar surface of R Doradus at 225 GHz.** The observations were taken using ALMA band 6 on 17 August 2023. The black ellipse in the panels indicates the average size of the stellar disc at 225 GHz.

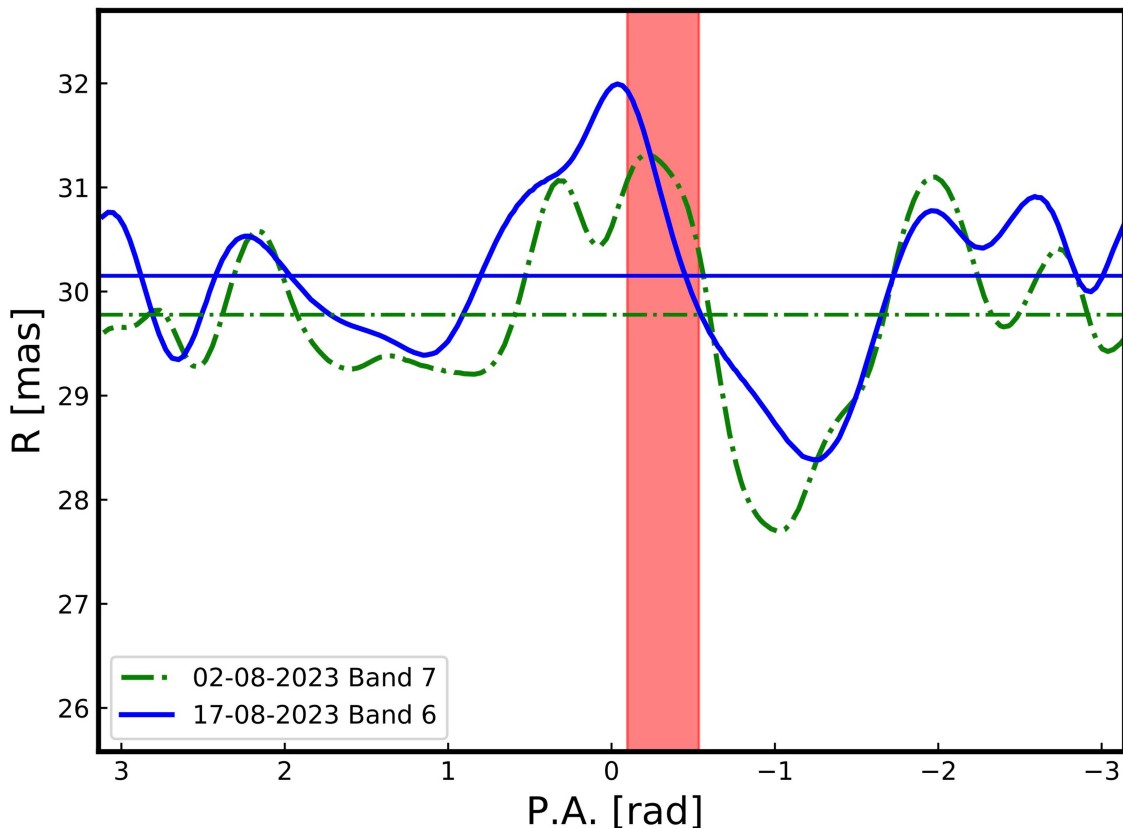

**Extended Data Fig. 4 | As Fig. 3a, the half-power radius of R Doradus as a function of the position angle.** The curves indicate the half-power radius measured for the final epochs of the ALMA Band 7 (338 GHz) observations compared with the ALMA Band 6 (225 GHz) observations taken 15 days later. The red bar is the granule size derived from the higher resolution Band 7 observations. The improved signal-to-noise in Band 6 compensates for the lower flux and larger beam so that the typical 1σ s.d. radius uncertainty is also of order 0.2 mas.

**Extended Data Table 1 | Observational details**

| Obs. date | Total/on-source time [hh:mm / hh:mm] | Min/max baseline length [m / m] | Beam size[1] [mas×mas, °] | rms noise[1] [mJy beam$^{-1}$] |
|---|---|---|---|---|
| 05-07-2023[2] | 1:41/0:36 | 113/9744 | $24.4 \times 13.3, -62.6$ | 0.14 |
| 08-07-2023[2] | 0:30/0:09 | 113/9237 | $15.6 \times 14.3, 57.4$ | 0.19 |
| 18-07-2023 | 0:39/0:11 | 230/15238 | $15.8 \times 8.8, 39.6$ | 0.38 |
| 27-07-2023 | 0:36/0:10 | 230/16196 | $12.2 \times 9.1, -23.4$ | 0.51 |
| 02-08-2023 | 1:41/0:36 | 230/16196 | $12.3 \times 8.3, -29.7$ | 0.31 |
| 17-08-2023[3] | 00:59/00:22 | 83/14852 | $18.7 \times 14.6, 34.4$ | 0.05 |

[1]Superuniform weighting.
[2]Data marked as *semi-pass* in ALMA data quality assessment.
[3]Observations in ALMA Band 6.