## [Peer Review File · Nature]

Manuscript Title: One month convection timescale on the surface of a giant evolved star

Reviewer Comments & Author Rebuttals

Reviewer Reports on the Initial Version:

Referees' comments:

Referee #1 (Remarks to the Author):

A. Summary of the key results

- The paper reports ALMA observations of the surface of the O-rich semiregular variable star R Dor. The images resolve the surface and show several structures which are interpreted as convective granules. Such kind of observations for stars other than the Sun are rare, and have been reported via aperture synthesis for the O-rich AGB π 1 Gruis, and red supergiants such as AZ Sgr (Norris et al 2012), V602 Car (Climent et al. 2020), and Betelgeuse (Kervella et al. A&A 609, A67 (2018)). The authors here determine the size of the convective patterns directly from the uv-plane using a custom-made code. They detect the presence of about 30 structures. They have different snapshots images within one month, and can observe directly via images the bubbling surface of the star, this is remarkable. The velocity of the surface motions is varying between -18 and $+20$ km s $^{-1}$ over a timescale of two weeks and structures exist for at least 3 weeks. The observations are in qualitative agreement with the theoretical predictions from stellar evolution.

B. Originality and significance: if not novel, please include reference

- There are a few works imaging stellar surfaces, mostly for red supergiants. Beside the π 1 Gru work cited by the authors there are also red supergiants such as AZ Sgr (Norris et al 2012), V602 Car (Climent et al. 2020), Betelgeuse (Kervella et al. A&A 609, A67 (2018), and CE Tau (Montargès et al. 2018).
- A velocity map from ALMA data for the stellar surface of Betelgeuse is derived in Kervella et al. A&A 609, A67 (2018). In the latter there is no time resolution (meaning different epochs), and the data are interpreted as related to convection in Jing-Ze Ma et al 2024 ApJL 962 L36. The AMBER observations of R Dor (same object presented in the work object of this review) presented in Ohnaka et al 2019 ApJ 883 89 show velocity maps with derived values of the same order of the one reported in the current work, however in the latter the authors interpret the motion as related to radiation pressure on dust grains. Convective movement as well as pulsation are introduced as alternative scenario in Ohnaka's work. Question to the authors: can the AMBER findings be re-interpreted in light of the new ALMA observations?
- Time resolved observations in terms of images are presented in the previously cited Norris paper, however these are data taken during few years. What I find really remarkable in the work object of this review is the number of observations within a relative short amount of time, and the evolution of the population of structures on the surface. I believe these observations deserve the publication in a high impact journal such as Nature.
- There have been a few works hinting to the timescale of convection on stellar surface being of the order of a month (CE Tau using aperture synthesis, Montargès et al. A&A 614, A12 (2018)), with few

cells surviving much longer (Norris, Climent works). As far as I am aware this is the first time we see it in terms of images, the first time for AGB stars (all the other works previously mentioned exception made for $\pi 1$ Gruis are for red supergiants) and it is very impressive. Because of this I recommend to consider these results for a Nature publication.

C. Data & methodology: validity of approach, quality of data, quality of presentation

- As mentioned previously the data are interpreted with a custom made code fitting the features in the Fourier space to avoid artefacts of the image reconstruction algorithm. This sounds as a reasonable approach. See point F. for a few questions concerning the effect of the beam.

D. Appropriate use of statistics and treatment of uncertainties

- No particular concerns.

E. Conclusions: robustness, validity, reliability

- I am quite confident about the robustness of the analysis, validity, and the reliability of the results presented. However in Sect. F. I will propose to address few points which in my opinion could enhance the impact of the paper.

F. Suggested improvements: experiments, data for possible revision

- The authors here interpret the observations as convective granules, however this is not always the case in the literature. Due to the limited angular resolution of observations there is a discussion in the community if the structures observed on the stellar surface are really convective granules (see Climent et al. 2020, A&A, 635, A160 their section 5.2 for a detailed discussion; AZ Sgr Norris et al 2012). Two scenarios are presented for V602 Car, while in the case of AZ Sgr the authors mention two populations of “convective features” (not granules!) with two different time scales.

I would like to ask the authors to consider including a short discussion (a couple of sentences) about the discrepancy of AGBs ($\pi 1$ Gru and R Dor) versus the red supergiants. How robust is the assumption that we are resolving individual convective granules? Why the structures detected on the surface of AGBs fit qualitatively the predictions from models (Tremblay, Trampedach, and Freytag), while for RSG there is considerable discrepancy? I believe this is an urgent question that should be addressed.

- Sect. 3.3 of Tremblay et al. A&A 557, A7 (2013) is about the characteristic life-time of the convective cells. Despite only 3 (5?) epochs of observations are available, I would encourage the authors if possible to consider expanding their discussion exploring the meaning of the derived time scale in the context of stellar evolution (better convection predictions across the HR-diagram), and not only in comparison with the 3D model predictions of Freytag, Liljegren & co-authors.

- How is the different beam size and orientation affecting the derived measurements? At page 4 the text mentions “There is a good correspondence between the epochs, with specifically the last two epochs showing very similar radial profiles.” How is this related to the similar interferometric beam?

- In Fig.4 it seems that the agreement between predictions and observations would improve if the grains measured are smaller. What is the impact of the angular resolution of the data on such plot? Is it possible that with higher angular resolution the measured “granules” would fit better? What is the effect of contrast on the images? In fact in Fourier plane the size and the contrast mix in the data.

G. References: appropriate credit to previous work?

- I am well aware that the space is limited, however I believe the authors should add references and a short discussion about the previous works on granulation size done on red supergiants. How do the newly presented results of R Dor fit in the literature scenario? (see point F.)

H. Clarity and context: lucidity of abstract/summary, appropriateness of abstract, introduction and conclusions

- I find the text overall well written and clear. I have one suggestion. The authors have 5 epochs, however the first two have significant lower angular resolution and are hardly comparable with the others (the size of the structures seem to be derived from only the other three images). I would propose to move the first two images in appendix. If the authors prefer to keep the five images (it is more impressive given the short time-lapse), I would refrain from making interpretation, or I would propose to phrase more strongly the word of caution in the comparison between the similar structures. If the structures are indeed the same it would be nice to see a PSD of the first two images.

Referee #2 (Remarks to the Author):

Dear editor, dear authors,

I am pleased to have the chance to participate as referee for the interesting manuscript entitled "The ever-changing surface of a bubbling giant star". The authors present a series of maps of the photosphere of the AGB star R Dor produced from high angular resolution ALMA data taken in bands 6 and 7 over a period of ~1 month. The maps show an irregular stellar disk that apparently changes over time. Moreover, the authors analyzed the data with simple methods to interpret the observations and derive key physical information about the behavior of the convective cells in the photosphere of R Dor. The text is well written, the reasonings clearly explained, and there is no unnecessary information. Despite most of the manuscript inspires confidence, I have found several major issues that prevent me to recommend this manuscript for publication in Nature at this moment. Hopefully, the authors will address these issues successfully and answer my questions. Below, you can find my major concerns first followed by some minor comments that the authors may want to consider.

Best wishes.

#####

MAJOR COMMENTS:

- Paladini et al. (2018) already published infrared images of the stellar disk of Pi1 Gru, which shows a

complex, irregular emission pattern. This pattern was claimed to result from granulation, as in the case of the maps included in the current manuscript. Hence, in spite of being scientifically valuable, the impact of the current manuscript is somewhat compromised. Nevertheless, the authors have also estimated the average gas velocity of the granules over the period of time covered by the monitoring. As far as I know, this velocity has not been ever measured but presenting this velocity is not the main goal of the manuscript. I would recommend to give more importance to the estimate of this velocity.

- I think that more works have to be cited regarding the analysis of convective cells in AGBs. Actually, the introduction is quite limited. I understand that Nature imposes very tight constraints of the text length but I feel that it is insufficient for non-astronomers (and perhaps also for astronomers not working on evolved stars). Considering the difficulty of observing these structures and the novelty of their study in AGBs, giving a wider view of the problem is necessary, in my opinion. From this point of view, Wittkowski et al. (2016, A&A, 587, A12) and Wittkowski et al. (2017, A&A, 601, A3) derived some conclusions about convective cells in evolved stars also using near-infrared interferometry. There are observational evidence of the effect of convective cells on the wind of AGBs in the vicinity of the stars. Velilla-Prieto et al. (2023, Nature, 617, 696) show these evidence for IRC+10216. The authors cite Vlemmings et al. (2017) in the text but not in the introduction. Using this work there to present other photospheric phenomena would be valuable. Any other work that I may have overlooked would be useful as well.

- The brightness distribution in the upper maps in Fig. 1 have a higher SNR than in the lower panels, mostly due to the lower angular resolution of the former. Considering that these maps are the basis of the whole manuscript, their presentation must be as flawless as possible. One can infer the presence of a structure from the upper maps but all the results are derived from the lower maps. These lower maps display a clear noise structure outside the star. It may be the color scale that is confusing me but I wonder if the structure in the stellar disk is real or it is an effect of noise. The text suggests that this structure is not noise but real, however this is not clearly supported. The elements of the structure in the stellar disk that are claimed to be granules must be detected with respect to the inter-granule space with a 5sigma level or higher (a 3sigma level might be acceptable but it should be supported by other findings). If this is the case, it is very important to make it clear in the text. In addition, it is not clear how noise affect the time evolution of the granulation pattern (section 2, par. 1). If the authors aim at presenting evidence of the time evolution of the granulation inside the stellar disk (not at the edge, which is more evident), including in their reasonings the effect of noise is necessary unless it can be neglected. How do the noise structure evolves from one observation to another?

MINOR COMMENTS:

- Abstract, line 15, "... images...":

This word is not commonly used in the field of mm interferometry but I understand that the public of Nature is much wider. However, an image suggest a picture and interferometric maps are not pictures but reconstructed images. I think that the word "reconstructed" should be appear in the abstract to prevent misunderstandings. Otherwise, they could be confused with images of, for instance, the JWST.

- Section 1, line 11, "...an AGB... of S-type AGB stars..."

As far as I know, the C/O ratio is not important regarding the formation of convective cells but I might be wrong. If so, justifying this statement with a reasoning or a reference would be required.

- Section 2, par. 1, line 4, "...using superuniform weighting...":

Including an explanation of what the superuniform weighting is or giving a reference that a potential reader of Nature with a limited knowledge of mm interferometry could understand is a good idea. A reference is given in the Methods section (difficult for beginners) but no explanation about this technique is included.

- Section 2, par. 1, line 5, "...produce images..." -> build/reconstruct images/maps

- Section 2, par. 1, line 6, "...resolution..." -> spatial/angular resolution

- Section 2, par. 1, line 16, "...significantly lower angular resolution...":

The angular resolution of the first epoch might be considered as significantly lower compared to the resolutions for the last three epochs but the second one has a similar angular resolution.

- Section 2, par. 1, line 19, "...a weaker feature... last two epochs":

This is very difficult to notice in the maps and, as I commented above, it is not clear what is the role of noise in this change. Apparently, it is true that there is an arc in the maps of the last two epochs in the place where the authors claim to have found this evolving feature but I cannot see the increase of brightness of the feature in the second and third epochs. This is in part because the angular resolution of the maps derived from the observations of the second epoch is worse than in the third

epoch and the feature the authors refer to can be smoothed and mixed with other close features. I suggest to equalize the angular resolutions. A comparison between maps produced with different synthesized PSFs can be only successfully made by experts in the field. Other reader will find difficult to understand the authors' conclusion based on the maps.

- Section 2, par. 1, line 23, "... to be brightening... the north-west":

This is difficult to believe given the noise of the map, as I mentioned above. It is crucial to demonstrate that the impact of noise is negligible on the shape of the structure. The authors state below that "The correspondence between independent epochs shows that the structures that we observe are indeed intrinsic to the star". They can be right about the overall shape of the structure but this does not imply that every detail can be trusted. This must be better supported.

- Fig. 1:

* I suggest to modify the color scale in order to increase the contrast between the so-called granules and the inter-granules emission. Perhaps, reducing the upper limit of this scale is enough. I am not usually in favor of using rainbow-like color scales but this time it may be justified because the structure to show is close to the maximum of the color scale. Another option is to add a new panel with a better choice of the color scale limits for the map of a given epoch in order to show the granulation in more detail.

* Plotting here the IR radius obtained by Ohnaka et al. (2019) would be useful. The authors nicely estimate the difference between the radius of the star in the mm and IR ranges in the Methods section. This is a very good place to mark where the IR emission comes from.

* Why not using the same round beam for the maps of the last three epochs to prevent elongation to modify the shape of the features in the stellar disk? Using a HPBW equal to the major axis of epoch 3 will be a good choice but using round beams with a HPBW equal to the $\sqrt{\text{major} \cdot \text{minor}}$ will work as well. I do not expect the increase of angular resolution along the direction of the major axis to introduce any significant artifact.

- Section 2, par. 2, line 10, "... $x=13.1 \pm 0.6$ mas...":

This size is derived from an analysis in the uv plane and it is thus not clear the exact meaning in the image plane. Would you assign this size to the FWHM of the smallest elements of the granulation, to the FWZI, or to the sigma parameter of a Gaussian, for instance?

- Fig. 2, caption, line 4, "...dots..." -> stars

- Fig. 2, caption, line 6, "... second bin..." -> dot at ~ 12.5 mas

- Fig. 2, caption, line 9, "...this corresponds to $x=0.72 \pm 0.05$ au":

Including the radius or the diameter of the star in AU would allow for comparisons.

- Section 2, par. 2, line 14, "The different scales... on our Sun":

The peaks you find can be related to the distances between what the authors call granules. Most of the granules are distributed close to the edge and the correlation between their positions will give the largest scale (~ 35 mas). The other peaks can be assigned to the correlations between the peaks in different positions. All the peaks in the PSD you show in Fig. 2 except the first one at ~ 13 mas may not imply larger structures but correlations between the positions of different granules. Thus, these correlations do not imply the presence of meso and supergranulation (considering the elements of the observed structure as granules; actually, the typical size of these elements, 0.72 AU, makes them more similar to supergranules than to granules, which show scales of 30 - 35 Mm and 0.5 - 2 Mm, respectively, or 2.0 - 2.5% and 0.03 - 0.1% the Sun's diameter; Rincon & Rieutord, 2018, Living Reviews in Solar Physics, 15, 6).

- Section 2, par. 5, line 3, "This is notably... near-infrared":

Giving the contrast would help understand this statement.

- Section 2, par. 5, line 9, "... this is sufficient... stellar atmosphere":

This statement needs a reference.

- Fig. 3:

* Plotting the uncertainties would allow to evaluate the effect of noise.

* Using degrees is clearer for the reader. In addition, the position angle is usually measured in degrees. Why using radians?

- Fig. 3, caption, line 6, "... estimated photospheric radius of 25.6 mas...":

At what wavelength was measured this radius?

- Section 3.1, par. 1, line 7, "... rotates fast for a giant star...":

Including here the typical rotation velocities expected for AGB stars would allow for comparisons.

- Section 3.1, par. 1, line 8, "... ± 0.1 km/s..." -> ± 0.1 km/s

- Section 3.1, par. 2, line 10, "... $D_{338\text{GHz}}=59.8 \pm 0.4$ mas...":

Despite it is included in the caption of Fig. 3, giving the condition to measure this diameter ($\tau=1$) also here would be helpful.

- Section 3.1, par. 2, line 12, "... $1.18 \pm 0.11 R_{\text{IR}}$ ":

Can the authors derive from the ratio $R_{338\text{GHz}}/R_{\text{IR}}$ any physical quantity or conclusion about the shell between the IR and mm radii?

- Section 3.2, par. 1, line 6, "... largest ALMA configurations...":

I imagine that these are configurations 9 and 10. Including them in the text (or in Table 1) would be helpful.

- Section 3.2, par. 1, line 9, "The first... using CASA v6.5.4.9":

What was the reason for the semi-pass assignment for the first epoch? I can see the problem for the second epoch in the next sentence. For the observations of this latter epoch, I assume that the requested angular resolution was not achieved. Am I right? If this was the only problem, it would be good to say it explicitly to prevent any doubt about the data quality.

- Section 3.2, par. 1, line 18, "... superuniform visibility weighting":

The explanation given in the cited paper is quite short. The use of this weighting is not common and including a brief explanation in the text (in addition to giving the reference) will be useful for the reader.

- Section 3.2, par. 1, line 22, "... of $\sim 2, 1.5, 1.5, 3, \dots$ five epochs...":

How much the angular resolution was improved after using the superuniform weighting (compared to the more regular uniform weighting)?

- Section 3.3, par. 1, line 14, "In addition, we also... from the star":

Can be only a position outside the star representative for the whole field of view (excluding the star)? What happens if another position is chosen? Is there any dependence with the distance from the center of the star or with the position angle?

- Fig. 5:

I miss the dispersion due to the axial average and the rms noise in both panels.

- Fig. 6:

Plotting a circle with the IR radius for comparison would be helpful.

- Fig. 6, caption, line 3, "The red bar... 7 observations":

I am unable to find the red bar. Is it there and my pdf viewer does not work properly, is this sentence a remainder of a previous version of the manuscript that you forgot to remove, or should this sentence be included in the caption of Fig. 7?

Author Rebuttals to Initial Comments:

We thank the referees for their good and constructive comments and for pointing out important references that were missed. We have attempted to address all the comments while also making some changes based on editorial comments and the requirements of the Nature structure. We address answers to the comments below. [Our answers are given in red.] We also include a figure to the reply to address some of the comments (although it is not intended to be included in the publication).

Specifically, we have added several references as requested by the referees. Based on editor comments we have also significantly shortened the boldface paragraph and any further introduction in order to comply with the Nature style requirements. For this, we have adopted the style template from a recently published paper. This included additional restructuring of the final paragraph. Note also that as requested, the figure legends from the main text have been moved to after the references. Finally, we have included, as csv tables, the data for a number of figures.

Referees' comments:

Referee #1 (Remarks to the Author):

- Question to the authors: can the AMBER findings [Ohnaka et al 2019 ApJ 883 89] be re-interpreted in light of the new ALMA observations?

Answer: The ALMA observations probe the atmosphere layer at about $1.18 R_*$ while the observations of Ohnaka et al. probe beyond $1.5 R_*$, where dust is being formed. At the location of the ALMA surface, no dust is expected to be present and the motions would not be the result of radiation pressure on dust grains. A re-interpretation of the Ohnaka et al. findings, in a layer outside the radius we present in the paper, is beyond the scope of the current paper.

- The authors here interpret the observations as convective granules, however this is not always the case in the literature. Due to the limited angular resolution of observations there is a discussion in the community if the structures observed on the stellar surface are really convective granules (see Climent et al. 2020, A&A, 635, A160 their section 5.2 for a detailed discussion; AZ Sgr Norris et al 2012). Two scenarios are presented for V602 Car, while in the case of AZ Sgr the authors mention two populations of “convective features” (not granules!) with two different time scales. I would like to ask the authors to consider including a short discussion (a couple of sentences) about the discrepancy of AGBs ($\pi 1$ Gru and R Dor) versus the red supergiants. How robust is the assumption that we are resolving individual convective granules? Why the structures detected on the surface of AGBs fit qualitatively the predictions from models (Tremblay, Trampedach, and Freytag), while for RSG there is considerable discrepancy? I believe this is an urgent question that should be addressed.

Answer: We thank the referee for the important suggestion and the very relevant references to add. While it remains an assumption that we're seeing the imprint of individual convective granules, we think it is indeed a robust assumption. As will be the case for all observations of stars other than the Sun, we will remain limited by resolution. This means that it is always possible that more compact features remain (in fact, we note the observations of more compact hotspots). However, if these instead are the imprint of

much smaller granules, then this would require a complete revision of our understanding of convection. At the moment, it is hard to point to other physical processes that would create the observed signatures across the stellar surface. It is true that that different convective features are suggested for the red supergiants. As also noted in our case, larger structures do exist, and as noted in some of the references, convection occurs on different scales. But particularly for red supergiants, it appears that Chiavassa et al. (2009) as noted in Norris et al. (2021) effectively conclude that the alpha parameter we introduce in eq. 1 is of order of 30-50. This means that there seems to be a difference of a factor of 2-3 in scales between the AGB stars and the supergiants. We have, within the constraints of the manuscript, added this observed difference at the end of the paper. However, based on only two sources so far, it is hard to draw further conclusions.

- Sect. 3.3 of Tremblay et al. A&A 557, A7 (2013) is about the characteristic life-time of the convective cells. Despite only 3 (5?) epochs of observations are available, I would encourage the authors if possible to consider expanding their discussion exploring the meaning of the derived time scale in the context of stellar evolution (better convection predictions across the HR-diagram), and not only in comparison with the 3D model predictions of Freytag, Liljegren & co-authors.

Answer: There are indeed further very interesting comparisons to make with the work of Tremblay et al. and we have added a few sentences of the timescale comparison that indicate that the AGB timescales might be less than that for the less evolved stars (with different range of surface gravity) investigated in this paper. This could indicate differences between convection properties.

- How is the different beam size and orientation affecting the derived measurements? At page 4 the text mentions “There is a good correspondence between the epochs, with specifically the last two epochs showing very similar radial profiles.” How is this related to the similar interferometric beam?

Answer: We have (also in answer to comments by the second referee) produced a separate version of Fig. 3, where the beams are also convolved to the largest one of the three epochs. We include the figure here but expect that it does not fit the paper (we add a sentence about the test). What can be seen is that the overall structure remains (as does the correspondence between the epochs and the Band 6 data) but that tangential smoothing of the features (now that the beam is larger than the scales found in the PSD) smooths out the velocity. This is the expected behaviour. When the resolution is below that of the smallest features, the smoothing effect is smaller. This is why we prefer to keep the current figure that presents the data without loss of information. It means that (at a PA of -50 of the disc, which is where the tangential smoothing in epoch 3 (the first of the three plotted) is largest, the derived velocity between epoch 3 and 4 might be slightly underestimated.

- In Fig.4 it seems that the agreement between predictions and observations would improve if the grains measured are smaller. What is the impact of the angular resolution of the data on such plot? Is it possible that with higher angular resolution the measured “granules” would fit better? What is the effect of contrast on the images? In fact in Fourier plane the size and the contrast mix in the data.

A version of paper figure 3 with the beams convolved to the largest beam of the three epochs plotted (epoch three of the five total epochs taken). As expected, smoothing the resolution to one larger than the structure scales reduces the radius variation and somewhat decreases the derived velocities. Because in the original version, the last two epochs have a resolution smaller than the derived smallest structure size, we prefer to use the original version which better represents the data.

Answer: It depends on which relation one favours. For example, as can be seen in the figure, Trampedach et al. (2013) predicts larger sizes. The impact of angular resolution (or more specifically, longer baselines) would be that smaller scale structures could become visible. In this particular observation, we cannot immediately rule out features with a size of <0.5 au. The observed features reported by us would then be chance alignments of convection granules or, more likely in that case, signatures of larger scale convection. However, this would also reduce the timescales involved (which are based on the size scale divided by velocity), which would further enhance the discrepancy between the predicted and observed timescales. Eventually, models would need to reproduce not just single convection scales but the entire power spectrum as well as the timescales. The current observations are the best constraints that can be achieved at the moment. The contrast is captured in the PSD and is well beyond anything that can be attributed to noise.

G. References: appropriate credit to previous work?

- I am well aware that the space is limited, however I believe the authors should add references and a short discussion about the previous works on granulation size done on red supergiants. How do the newly presented results of R Dor fit in the literature scenario? (see point F.)

Answer: A few lines have been added (as well as references).

H. Clarity and context: lucidity of abstract/summary, appropriateness of abstract, introduction and conclusions

- I find the text overall well written and clear. I have one suggestion. The authors have 5 epochs, however the first two have significant lower angular resolution and are hardly comparable with the others (the size of the structures seem to be derived from only the other three images). I would propose to move the first two images in appendix. If the authors prefer to keep the five images (it is more impressive given the short time-lapse), I would refrain from making interpretation, or I would propose to phrase more strongly the word of caution in the comparison between the similar structures. If the structures are indeed the same it would be nice to see a PSD of the first two images.

Answer: We agree with the referee and have made the suggested change. We have also, both in response to the comment and for manuscript length constraints, removed most of the discussion based on visual interpretations.

Referee #2 (Remarks to the Author):

MAJOR COMMENTS:

- Paladini et al. (2018) already published infrared images of the stellar disk of Pi1 Gru, which shows a complex, irregular emission pattern. This pattern was claimed to result from granulation, as in the case of the maps included in the current manuscript. Hence, in spite of being scientifically valuable, the impact of the current manuscript is somewhat compromised. Nevertheless, the authors have also estimated the average gas velocity of the granules over the period of time covered by the monitoring. As far as I know, this velocity has not been ever measured but presenting this velocity is not the main goal of the manuscript. I would recommend to give more importance to the estimate of this velocity.

Answer: We thank the referee for highlighting that the velocity determination was not sufficiently prominent in the paper. It was certainly intended to be one of the main important points. Within the constraints on the manuscript length we hope this is now better visible, given also by the additional comparison with the result on the timescales mentioned by referee #1.

- I think that more works have to be cited regarding the analysis of convective cells in AGBs. Actually, the introduction is quite limited. I understand that Nature imposes very tight constraints of the text length but I feel that it is insufficient for non-astronomers (and perhaps also for astronomers not working on evolved stars). Considering the difficulty of observing these structures and the novelty of their study in AGBs, giving a wider view of the problem is necessary, in my opinion. From this point of view, Wittkowski et al. (2016, A&A, 587, A12) and Wittkowski et al. (2017, A&A, 601, A3) derived some conclusions about convective cells in evolved stars also using near-infrared interferometry. There are observational evidence of the effect of convective cells on the wind of AGBs in the vicinity of the stars. Velilla-Prieto et al. (2023, Nature, 617, 696) show these evidence for IRC+10216. The authors cite Vlemmings et al. (2017) in the text but not in the introduction.

Using this work there to present other photospheric phenomena would be valuable. Any other work that I may have overlooked would be useful as well.

Answer: We have added references to these papers (and those mentioned by referee #1).

- The brightness distribution in the upper maps in Fig. 1 have a higher SNR than in the lower panels, mostly due to the lower angular resolution of the former. Considering that these maps are the basis of the whole manuscript, their presentation must be as flawless as possible. One can infer the presence of a structure from the upper maps but all the results are derived from the lower maps. These lower maps display a clear noise structure outside the star. It may be the color scale that is confusing me but I wonder if the structure in the stellar disk is real or it is an effect of noise. The text suggests that this structure is not noise but real, however this is not clearly supported. The elements of the structure in the stellar disk that are claimed to be granules must be detected with respect to the inter-granule space with a 5sigma level or higher (a 3sigma level might be acceptable but it should be supported by other findings). If this is the case, it is very important to make it clear in the text. In addition, it is not clear how noise affect the time evolution of the granulation pattern (section 2, par. 1). If the authors aim at presenting evidence of the time evolution of the granulation inside the stellar disk (not at the edge, which is more evident), including in their reasonings the effect of noise is necessary unless it can be neglected. How do the noise structure evolves from one observation to another?

Answer: Based on comments from referee #1 we focus now on the last three epochs in the first figure. We also reduce the text on the visual comparison because the velocity (and velocity derived timescale) and the visibility analysis are the core of the work. The reconstructed images serve primarily as illustration. We experimented with a variety of color scales. In the end, we decided to instead add contours that present the 4, 5, and 6 sigma significance of the features on top of the average disc emission. We do this only for the highest angular resolution images considering the other images do not have sufficient angular resolution to resolve the features. As can be seen, the features peak at close to 7 sigma with several features visible above 5 sigma. Considering the majority of the features are significantly above the effect of noise, the noise effect on the stellar disc structure is small. However, also for manuscript length considerations we removed the biggest part describing the observed features, as the timescale is primarily derived from the velocity measurement.

MINOR COMMENTS:

- Abstract, line 15, "... images...":

This word is not commonly used in the field of mm interferometry but I understand that the public of Nature is much wider. However, an image suggest a picture and interferometric maps are not pictures but reconstructed images. I think that the word "reconstructed" should be appear in the abstract to prevent misunderstandings. Otherwise, they could be confused with images of, for instance, the JWST.

Answer: Although 'interferometric images' was meant to convey the meaning, we have added 'reconstructed'.

- Section 1, line 11, "...an AGB... of S-type AGB stars..."

As far as I know, the C/O ratio is not important regarding the formation of convective cells but I might be wrong. If so, justifying this statement with a reasoning or a reference would be required.

Answer: It is indeed unlikely that the C/O ratio plays a role. Also for manuscript length purposes we have removed this qualifier.

- Section 2, par. 1, line 4, "...using superuniform weighting...":

Including an explanation of what the superuniform weighting is or giving a reference that a potential reader of Nature with a limited knowledge of mm interferometry could understand is a good idea. A reference is given in the Methods section (difficult for beginners) but no explanation about this technique is included.

Answer: We have added a line in the methods description, but interferometry and its interpretation will always remain difficult for beginners and for the interested reader who wants to know more background, the references (and references therein) will need to be followed.

- Section 2, par. 1, line 5, "...produce images..." -> build/reconstruct images/maps

- Section 2, par. 1, line 6, "...resolution..." -> spatial/angular resolution

- - Section 2, par. 1, line 16, "...significantly lower angular resolution...":

The angular resolution of the first epoch might be considered as significantly lower compared to the resolutions for the last three epochs but the second one has a similar angular resolution.

Answers: all changes made.

- Section 2, par. 1, line 19, "...a weaker feature... last two epochs":

This is very difficult to notice in the maps and, as I commented above, it is not clear what is the role of noise in this change. Apparently, it is true that there is an arc in the maps of the last two epochs in the place where the authors claim to have found this evolving feature but I cannot see the increase of brightness of the feature in the second and third epochs. This is in part because the angular resolution of the maps derived from the observations of the second epoch is worse than in the third epoch and the feature the authors refer to can be smoothed and mixed with other close features. I suggest to equalize the angular resolutions. A comparison between maps produced with different synthesized PSFs can be only successfully made by experts in the field. Other reader will find difficult to understand the authors' conclusion based on the maps.

Answer: As noted above, we have removed this discussion. We have also experimented with changing the resolutions but this either introduces significant information loss for the highest resolution images or introduces scales not available in the lower resolution images (overresolving an image reconstructed using superuniform weighing should not be done).

- Section 2, par. 1, line 23, "... to be brightening... the north-west":

This is difficult to believe given the noise of the map, as I mentioned above. It is crucial to demonstrate that the impact of noise is negligible on the shape of the structure. The authors state below that "The correspondence between independent epochs shows that the structures that we observe are indeed intrinsic to the star". They can be right about the overall shape of the structure but this does not imply that every detail can be trusted. This must be better supported.

Answer: This part has been removed and the comparison is based on the radius plots where we have done a lower resolution comparison (see the included figure above).

- Fig. 1:

* I suggest to modify the color scale in order to increase the contrast between the so-called granules and the inter-granules emission. Perhaps, reducing the upper limit of this scale is enough. I am not usually in favor of using rainbow-like color scales but this time it may be justified because the structure to show is close to the maximum of the color scale. Another option is to add a new panel with a better choice of the color scale limits for the map of a given epoch in order to show the granulation in more detail.

Answer: We experimented with various color scales, but in the end decided that adding significance contours conveys the information best.

* Plotting here the IR radius obtained by Ohnaka et al. (2019) would be useful. The authors nicely estimate the difference between the radius of the star in the mm and IR ranges in the Methods section. This is a very good place to mark where the IR emission comes from.

Answer: We prefer to leave this out since, with the addition of the contours, the images are already sufficiently crowded.

* Why not using the same round beam for the maps of the last three epochs to prevent elongation to modify the shape of the features in the stellar disk? Using a HPBW equal to the major axis of epoch 3 will be a good choice but using round beams with a HPBW equal to the $\sqrt{\text{major} \times \text{minor}}$ will work as well. I do not expect the increase of angular resolution along the direction of the major axis to introduce any significant artifact.

Answer: As noted, lowering the resolution below that produced using superuniform weighing actually does introduce unphysical structures so we have decided against this.

- Section 2, par. 2, line 10, "... $x=13.1 \mu\text{m}$ 0.6 mas...":

This size is derived from an analysis in the uv plane and it is thus not clear the exact meaning in the image plane. Would you assign this size to the FWHM of the smallest elements of the granulation, to the FWZI, or to the sigma parameter of a Gaussian, for instance?

Answer: The power spectrum characterises the size of fluctuations in a cross-correlation. The exact interpretation of the size depends on the type of structure (often for example compared to CMB maps, ISM turbulence). In the end, the comparison needs to come with the power spectrum determined from the models that should be reproduced. The value reported is the characteristic size scale that contains most of the flux and will most closely correspond to the FWZI after the disc is subtracted. But see for example Aime

1978A&A....67....1A on the morphological interpretation. Considering the well established use of the PSD we refrain from expanding the description.

- Fig. 2, caption, line 4, "...dots..." -> stars
- Fig. 2, caption, line 6, "... second bin..." -> dot at $\sim 12.5\text{mas}$
- Fig. 2, caption, line 9, "...this corresponds to $x=0.72\text{pm}$ 0.05au ":

Including the radius or the diameter of the star in AU would allow for comparisons.

Answers: these changes have been made.

- Section 2, par. 2, line 14, "The different scales... on our Sun":

The peaks you find can be related to the distances between what the authors call granules. Most of the granules are distributed close to the edge and the correlation between their positions will give the largest scale ($\sim 35\text{mas}$). The other peaks can be assigned to the correlations between the peaks in different positions. All the peaks in the PSD you show in Fig. 2 except the first one at $\sim 13\text{mas}$ may not imply larger structures but correlations between the positions of different granules. Thus, these correlations do not imply the presence of meso and supergranulation (considering the elements of the observed structure as granules; actually, the typical size of these elements, 0.72AU , makes them more similar to supergranules than to granules, which show scales of $30\text{-}35\text{Mm}$ and $0.5\text{-}2\text{Mm}$, respectively, or $2.0\text{-}2.5\%$ and $0.03\text{-}0.1\%$ the Sun's diameter; Rincon & Rieutord, 2018, Living Reviews in Solar Physics, 15, 6).

Answer: We have extended the discussion on this point.

- Section 2, par. 5, line 3, "This is notably... near-infrared":

Giving the contrast would help understand this statement.

Answer: We have added the approximate value from the reference.

- Section 2, par. 5, line 9, "... this is sufficient... stellar atmosphere":

This statement needs a reference.

Answer: We have added a reference.

- Fig. 3:

* Plotting the uncertainties would allow to evaluate the effect of noise.

Answer: Bars indicating the uncertainties are added.

* Using degrees is clearer for the reader. In addition, the position angle is usually measured in degrees. Why using radians?

Answer: We have changed it to degrees.

- Fig. 3, caption, line 6, "... estimated photospheric radius of 25.6mas ...":

At what wavelength was measured this radius?

Answer: We have added the wavelength from the reference.

- Section 3.1, par. 1, line 7, "... rotates fast for a giant star...":

Including here the typical rotation velocities expected for AGB stars would allow for comparisons.

Answer: We have added the approximate value from the reference.

- Section 3.1, par. 1, line 8, "... ± 0.1 km/s..." -> ± 0.1 km/s

- Section 3.1, par. 2, line 10, "... $D_{338\text{GHz}} = 59.8 \pm 0.4$ mas...":

Despite it is included in the caption of Fig. 3, giving the condition to measure this diameter ($\tau=1$) also here would be helpful.

Answers: done

- Section 3.1, par. 2, line 12, "... $1.18 \pm 0.11 R_{\text{IR}}$ ":

Can the authors derive from the ratio $R_{338\text{GHz}}/R_{\text{IR}}$ any physical quantity or conclusion about the shell between the IR and mm radii?

Answers: This ratio, when including further frequencies, can be used to obtain constraints on the density, temperature and ionisation distributions of the gas. For this particular data this will be part of a separate paper, previously this was done for R Dor in Vlemmings et al. (2019). It is beyond the scope of the paper.

- Section 3.2, par. 1, line 6, "... largest ALMA configurations...":

I imagine that these are configurations 9 and 10. Including them in the text (or in Table 1) would be helpful.

Answer: added

- Section 3.2, par. 1, line 9, "The first... using CASA v6.5.4.9":

What was the reason for the semi-pass assignment for the first epoch? I can see the problem for the second epoch in the next sentence. For the observations of this latter epoch, I assume that the requested angular resolution was not achieved. Am I right? If this was the only problem, it would be good to say it explicitly to prevent any doubt about the data quality.

Answer: added

- Section 3.2, par. 1, line 18, "... superuniform visibility weighting":

The explanation given in the cited paper is quite short. The use of this weighting is not common and including a brief explanation in the text (in addition to giving the reference) will be useful for the reader.

Answer: a very short additional explanation is added, but for a more detailed description the interested reader needs to follow the reference.

- Section 3.2, par. 1, line 22, "... of $\sim 2, 1.5, 1.5, 3, \dots$ five epochs...":

How much the angular resolution was improved after using the superuniform weighting (compared to the more regular uniform weighting)?

Answer: added

- Section 3.3, par. 1, line 14, "In addition, we also... from the star":

Can be only a position outside the star representative for the whole field of view (excluding the star)? What happens if another position is chosen? Is there any dependence with the distance from the center of the star or with the position angle?

Answer: There is no dependence on position angle. As was noted, the offset needs to be sufficiently away from the star to avoid a contribution from the true signal.

- Fig. 5:

I miss the dispersion due to the axial average and the rms noise in both panels.

Answer: Error bars have now been added for the fractional residual. For the azimuthal averaging, the error bars turn out to be barely larger than the width of the line, and adding them made the figure confusing. The largest value of the error is added in the caption.

- Fig. 6:

Plotting a circle with the IR radius for comparison would be helpful.

Answer: We feel that the textual comparison is sufficient without further complicating the images.

- Fig. 6, caption, line 3, "The red bar... 7 observations":

I am unable to find the red bar. Is it there and my pdf viewer does not work properly, is this sentence a remainder of a previous version of the manuscript that you forgot to remove, or should this sentence be included in the caption of Fig. 7?

Answer: Thank you for pointing this out. Yes, this should have been in the caption of the next figure. Moved.

Reviewer Reports on the First Revision:

Referee #2 (Remarks to the Author):

A. Summary of the key results

The authors present the first description in the mm range carried out with ALMA in its most extended configurations of what is apparently the convection structure in the extended atmosphere of an AGB star, taking the semi-regular star R Doradus as target. Apart of showing this structure noticed inside the stellar disk, the authors also derive the average velocity of the gas in the convective cells detected in the limb of the star, which moves essentially in the plane of the sky. The authors analyze these results using parametric formulas related to the currently accepted theory of convection in red giants. A deeper analysis of the data can be done but it is out of the scope of this paper, which aims at presenting the observations and a qualitative interpretation.

B. Originality and significance: if not novel, please include reference

The first report of the convective structure of an AGB star was done a few years ago (Paladini et al., 2018, Nature, 553, 310; cited in the manuscript) but this is the first time, to my knowledge, that the velocity of the gas in these cells is estimated. This represents an important step to be taken in the analysis and comprehension of the convective mechanisms at work in the atmospheres of evolved stars and can be compared with predictions of complex numerical models (that certainly will be done in the future). In the new version of the manuscript, the authors give more importance to the determination of the velocity of the gas in the convective cells than in the first one, as it was suggested.

C. Data & methodology: validity of approach, quality of data, quality of presentation

The presented data is of high quality and has been reduced and analyzed by experts in the matter (the authors themselves and the staff of the Nordic ARC). The data presentation was good in the first version of the manuscript and has been improved since then based on our comments and suggestions. The methodology is right and the reasonings and choices of the authors make sense. However, I still have a few more comments and questions:

- p. 3, par. 1, line 8: "...derive the average velocity profile..."

How did the authors do it? Did they linear fit the three radii for each PA or did they use another method?

- p. 3, par. 1, line 13: "...shocks..."

Something that I find unsettling when the authors discuss about shocks is that it is not clear to me what a shock is in this scenario. Are we witnessing the shocks themselves or the effect of the shocks on the external layers of the (extended) atmosphere? Where are the shocks expected to occur, in the extended atmosphere or in the photosphere, if we consider this latter layer as that observed in the near-IR? Can the authors provide a reference to solve this questions or can they briefly explain what they understand for a shock? Why is it necessary to consider the existence of shocks and it is not possible that the observed spots are only hotter matter extracted from the insides of the star by the convective cells?

D. Appropriate use of statistics and treatment of uncertainties

The treatment of uncertainties is well done. It was fine in the first version of the manuscript and the authors improved them following our comments. However, I suggest to give the errors of the contrasts presented in p. 5, par. 2, line 2. It is not clear if the difference between these quantities is statistically significant (see F). The data analysis method followed by the authors could be demonstrated to be useful to show significant changes of the structure found inside the stellar disk over time which are different from those related to the structure detected in the limb of the star (Fig. 3). Moreover, short term variations of the light curve might be evidenced as well.

E. Conclusions: robustness, validity, reliability

The method followed by the authors to analyze the data allows them to derive reliable results and sound conclusions derived from them. Their reasonings seem accurate in general and are based on previous research. Only the use of the word "granule" to refer to the observed spots could be a subject of discussion. Considering the different temperatures, gravity, and size that characterize the Sun and RGB/AGB stars and the non-linearity of convection, the assignment of the observed compact structures as granules could be questioned. The connection between these structures and the solar granules is straightforward but can hide a bias. In the end, the compact structures that can be considered as granules are point-like in the ALMA observations and we can only be sure about the fact that they are separated by regions with a lower brightness temperature. Almost nothing can be said about the physical mechanism behind them. Despite the sentence "The different scales might... several granules" in p. 2, par. 2, line 11 aims at establishing this possible connection leaving some space for disagreements, I suggest to include an additional sentence that explains that establishing this connection in a solid way requires further investigation due to the complexity of the problem.

F. Suggested improvements: experiments, data for possible revision

- p. 5, par. 2, line 7: "We find... 1500K"

This T_b increase is related to a typical granule with respect to the average T_b over the stellar disk. Above, the authors provide the contrast between the granules and the average stellar disk and it changes from one epoch to another. Does the total stellar flux vary with time during the period spanned by the monitoring? Is the light curve essentially flat (or shows a linear dependence with time) or can the authors infer noticeable variations with time? If the light curve is flat/linear over time, can the differences between the contrasts presented above be significant? If the light curve is not flat/linear with time, is the variation in agreement with what is expected for the long period component of the stellar light curve or can they detect short term changes?

G. References: appropriate credit to previous work?

Perhaps other references can be given to support the author's reasonings and results (see my other comments) but I think those already included are usually well chosen and placed within the manuscript. There is an exception related to which I would like the authors to explain the reason why it is included in this particular sentence:

- p. 3, par. 1, line 16: "...vlsr=-14 to +26 km/s..."

Perhaps I overlooked this result but I am unable to find it in Vlemmings et al., 2018, A&A, 613, L4. Did the authors reinterpret the results presented in this paper to derive this velocity range? Where was that velocity range obtained from? From the observations that are being presented in the current manuscript or from Vlemmings et al. (2018)?

The measurement of the radial velocities presented in the current work is supposed to be the first one ever (at least in evolved stars; I do not know in other stellar types). However, this sentence apparently suggests the opposite if the velocity range was derived by Vlemmings et al. (2018).

H. Clarity and context: lucidity of abstract/summary, appropriateness of abstract, introduction and conclusions

- Abstract, line 10: "The images... on the stellar surface"

I think the transition from this sentence to the next one is somewhat abrupt for an abstract. I suggest to very briefly describe the detected structure to let the reader know how the dominant structure looks like before giving the typical size of the spots/granules/compact structures.

- p. 2, par. 1, line 9: "... molecular opacity... other wavelengths"

I agree with the mild effect of dust in the (sub)mm but not concerning the molecular opacity, which can strongly opaque the photosphere depending on the observed frequency, as it occurs at other wavelengths. The point is to select an observing frequency free of absorption or emission of

molecular lines. In the end, spectral resolution and SNR are the important parameters on this matter. (Sub)mm interferometry is a very good technique to observe the photosphere but not the only one, mostly for a star as R Dor, whose mass-loss rate is low.

- p. 2, par. 1, line 15: "Supplementary material"

Perhaps it is my fault but I feel that the use of "supplementary material" is somewhat confusing. Using "Methods" (which is the name of the section) is a better choice. What subsection do the authors refer to? I find other similar cases throughout the manuscript.

- p. 2, par. 1, line 16: "...and are induced by surface convection"

The correspondence between several epochs can lead us to conclude that the structure is real, indeed. However, any thorough conclusion about the nature of this structure requires further reasonings. I suggest to rephrase the sentence to "...and are (very) likely/probably induced by surface convection" or something similar. Another possibility is to move this part of the sentence to the end of the next one: "...we observe at the surface have a typical lifetime of at least three weeks, which is compatible with convection". Either case, a reference is needed to justify the correspondence of this time varying structure with what we know as convection and convective cells in stars.

- p. 2, par. 2, line 14: "... granules" -> "... independent/uncorrelated granules"

- p. 3, par. 1, line 1: "The structures on the star..." -> "The structures inside the stellar disc..." (?)

- p. 3, par. 1, line 7: "... difference in radius..." -> "...difference in radius at every PA/depending on the position angle..."

- p. 3, par. 1, line 10: "...average velocity is low..."

"Low" here stands for low velocity modulus, right?

- p. 4, par. 1, line 3: "...the gravity field of the star"

I suggest to include here an explicit mention to the accepted gas acceleration mechanism based on radiation pressure on dust grains. Considering that AGBs are surrounded by an expanding envelope, including this clarification makes sense. Many non-specialists will read the paper if it is finally accepted for publication.

- p. 5, par. 1, line 8: "...emperical formula from [22]"

Including the number of this formula in [22] would help. Is it Eq. 16?

- p. 5, par. 2, line 3: "... (Fig. 2)"

Is it Fig. 1?

- p. 6, par. 2, line 2: "...but reveal a difference... red supergiants"

This is what the authors state at the end of the previous page, isn't it?

- p. 10, caption of Fig. 2, line 3: "...spline..."

What degree did the authors use? Cubic spline fitting is a typical tool to be used but other degrees can be used as well.

- p. 10, caption of Fig. 2, line 3: "...s.d..." -> "...standard deviation..."

I wonder if it is necessary to include it.

- p. 12, par. 1, line 16: "...Fig. 1" -> "...Extended Data Fig. 1" (?)

- p. 12, par. 1, line 24: "...supplementary material 1"

Can you give the section (or subsection)?

- p. 13, "Stellar disc profile", line 6: "...Fig. 2"

I understand that this Fig. 2 is the Fig. 2 of the Methods section (Extended Data Fig. 2) rather than that figure in the main body of the paper. I guess this would be solved during the edition process. Otherwise, I recommend to do it because it is confusing.

Author Rebuttals to First Revision:

Referee reply 2

Regarding the referee comments, we here reply to those of Referee 2, after Referee 1 suggested no further changes. We thank both referees for in depth reading of the manuscript.

p. 3, par. 1, line 8: "...derive the average velocity profile..."

How did the authors do it? Did they linear fit the three radii for each PA or did they use another method?

We did exactly as described and took the difference between subsequent epochs for each PA. Hence this is an average velocity between the epochs (which means that there are two velocity curves). We performed no fits.

- p. 3, par. 1, line 13: "...shocks..."

Something that I find unsettling when the authors discuss about shocks is that it is not clear to me what a shock is in this scenario. Are we witnessing the shocks themselves or the effect of the shocks on the external layers of the (extended) atmosphere? Where are the shocks expected to occur, in the extended atmosphere or in the photosphere, if we consider this latter layer as that observed in the near-IR? Can the authors provide a reference to solve this questions or can they briefly explain what they understand for a shock? Why is it necessary to consider the existence of shocks and it is not possible that the observed spots are only hotter matter extracted from the insides of the star by the convective cells?

As noted in our reply to the editor, these are really shocks. The observations indicate a sharp change of conditions and the velocities exceed the local sounds speed of 6 km/s, making them supersonic.

The treatment of uncertainties is well done. It was fine in the first version of the manuscript and the authors improved them following our comments. However, I suggest to give the errors of the contrasts presented in p. 5, par. 2, line 2. It is not clear if the difference between these quantities is statistically significant (see F).

We added uncertainties that follow directly from the PSD in figure 2.

Only the use of the word "granule" to refer to the observed spots could be a subject of discussion. Considering the different temperatures, gravity, and size that characterize the Sun and RGB/AGB stars and the non-linearity of convection, the assignment of the observed compact structures as granules could be questioned. The connection between these structures and the solar granules is straightforward but can hide a bias. In the end, the compact structures that can be considered as granules are point-like in the ALMA observations and we can only be sure about the fact that they are separated by regions with a lower brightness temperature. Almost nothing can be said about the physical mechanism behind them. Despite the sentence "The different scales might... several granules" in p. 2, par. 2, line 11 aims at establishing this possible connection leaving some space for disagreements, I suggest to include an additional sentence that explains that establishing this connection in a solid way requires further investigation due to the complexity of the problem.

We have followed the suggestion of the editor (and reworded some sentences following the comments below).

- p. 5, par. 2, line 7: "We find... 1500K"

This Tb increase is related to a typical granule with respect to the average Tb over the stellar disk. Above, the authors provide the contrast between the granules and the average stellar disk and it changes from one epoch to another. Does the total stellar flux vary with time during the period spanned by the monitoring? Is the light curve essentially flat (or shows a linear dependence with time) or can the authors infer noticeable variations with time? If the light curve is flat/linear over time, can be the differences between the contrasts presented above significant? If light curve is not flat/linear with time, is the variation in agreement with what is expected for the long period component of the stellar light curve or can they detect short term changes?

This is indeed with respect to the average Tb over the disc as derived from the PSD (so fluxes converted to brightness temperature). While relative values are accurate, the absolute values per epoch suffer from absolute

ALMA flux calibration errors of the order of 10% that make a comparison of absolute numbers between the epochs difficult.

- p. 3, par. 1, line 16: "...vlsr=-14 to +26 km/s..."

Perhaps I overlooked this result but I am unable to find it in Vlemmings et al., 2018, A&A, 613, L4. Did the authors reinterpret the results presented in this paper to derive this velocity range? Where was that velocity range obtained from? From the observations that are being presented in the current manuscript or from Vlemmings et al. (2018)?

The measurement of the radial velocities presented in the current work is supposed to be the first one ever (at least in evolved stars; I do not know in other stellar types). However, this sentence apparently suggests the opposite if the velocity range was derived by Vlemmings et al. (2018).

See our reply to the editor. We have removed this reference, since it confused the issue.

Abstract, line 10: "The images... on the stellar surface"

I think the transition from this sentence to the next one is somewhat abrupt for an abstract. I suggest to very briefly describe the detected structure to let the reader know how the dominant structure looks like before giving the typical size of the spots/granules/compact structures.

We have tried to make the transition somewhat better while maintaining the Nature constraints.

- p. 2, par. 1, line 9: "... molecular opacity... other wavelengths"

I agree with the mild effect of dust in the (sub)mm but not concerning the molecular opacity, which can strongly opaque the photosphere depending on the observed frequency, as it occurs at other wavelengths. The point is to select an observing frequency free of absorption or emission of molecular lines. In the end, spectral resolution and SNR are the important parameters on this matter. (Sub)mm interferometry is a very good technique to observe the photosphere but not the only one, mostly for a star as R Dor, whose mass-loss rate is low.

It was not the intention to make it appear this was the only method, but in the submm regime (and with the spectral resolution provided by ALMA) it is indeed a lot easier to find a line free region. But expressing this in the sentence with all the caveats would needlessly complicate the text.

- p. 2, par. 1, line 15: "Supplementary material"

Perhaps it is my fault but I feel that the use of "supplementary material" is somewhat confusing. Using "Methods" (which is the name of the section) is a better choice. What subsection do the authors refer to? I find other similar cases throughout the manuscript.

This should indeed have read the 'Methods' section.

- p. 2, par. 1, line 16: "...and are induced by surface convection"

The correspondence between several epochs can lead us to conclude that the structure is real, indeed. However, any thorough conclusion about the nature of this structure requires further reasonings. I suggest to rephrase the sentence to "...and are (very) likely/probably induced by surface convection" or something similar. Another possibility is to move this part of the sentence to the end of the next one: "...we observe at the surface have a typical lifetime of at least three weeks, which is compatible with convection". Either case, a reference is needed to justify the correspondence of this time varying structure with what we know as convection and convective cells in stars.

The 'very likely' qualifier is added and a references to the models that include timescales is added (although it followed in the next section).

- p. 2, par. 2, line 14: "... granules" -> "... independent/uncorrelated granules"

done

- p. 3, par. 1, line 1: "The structures on the star..." -> "The structures inside the stellar disc..." (?)

done

- p. 3, par. 1, line 7: "... difference in radius..." -> "...difference in radius at every PA/depending on the position

angle..."

done

- p. 3, par. 1, line 10: "...average velocity is low..."

"Low" here stands for low velocity modulus, right?

This was changed to 'small' as low was indeed wrong in this context.

- p. 4, par. 1, line 3: "...the gravity field of the star"

I suggest to include here an explicit mention to the accepted gas acceleration mechanism based on radiation pressure on dust grains. Considering that AGBs are surrounded by an expanding envelope, including this clarification makes sense. Many non-specialists will read the paper if it is finally accepted for publication.

We have added the reference to one of the Freytag and Höffner papers that was already cited in another context to avoid going over the limit for references.

- p. 5, par. 1, line 8: "...empirical formula from [22]"

Including the number of this formula in [22] would help. Is it Eq. 16?

Indeed. We added the number.

- p. 5, par. 2, line 3: "... (Fig. 2)"

Is it Fig. 1?

No, the contrast was calculated from the total power in the granules and the stellar disc from the PSD.

- p. 6, par. 2, line 2: "...but reveal a difference... red supergiants"

This is what the authors state at the end of the previous page, isn't it?

Yes. This is the very short part of the final discussion.

- p. 10, caption of Fig. 2, line 3: "...spline..."

What degree did the authors use? Cubic spline fitting is a typical tool to be used but other degrees can be used as well.

A cubic splint indeed. This was added.

- p. 10, caption of Fig. 2, line 3: "...s.d..." -> "...standard deviation..."

I wonder if it is necessary to include it.

We followed Nature guidelines that stated it needed to be included.

- p. 12, par. 1, line 16: "...Fig. 1" -> "...Extended Data Fig. 1" (?)

done

- p. 12, par. 1, line 24: "...supplementary material 1"

Can you give the section (or subsection)?

For some reason, the labelling did not work properly. It should now be correct.

- p. 13, "Stellar disc profile", line 6: "...Fig. 2"

I understand that this Fig. 2 is the Fig. 2 of the Methods section (Extended Data Fig. 2) rather than that figure in the main body of the paper. I guess this would be solved during the edition process. Otherwise, I recommend to do it because it is confusing.

done